# *BRAF* V600E Mutation in Ameloblastoma: A Systematic Review and Meta-Analysis

**DOI:** 10.3390/cancers14225593

**Published:** 2022-11-14

**Authors:** Mohd Nazzary Mamat @ Yusof, Ewe Seng Ch’ng, Nawal Radhiah Abdul Rahman

**Affiliations:** 1Department of Clinical Medicine, Advanced Medical and Dental Institute (AMDI), Universiti Sains Malaysia, Kepala Batas 13200, Malaysia; 2Department of Obstetrics and Gynaecology, Universiti Kebangsaan Malaysia Medical Centre, Kuala Lumpur 56000, Malaysia; 3Department of Dental Science, Advanced Medical and Dental Institute (AMDI), Universiti Sains Malaysia, Kepala Batas 13200, Malaysia

**Keywords:** ameloblastoma, odontogenic tumour, proto-oncogene proteins B-Raf, *BRAF* V600E, clinicopathological features

## Abstract

**Simple Summary:**

Ameloblastoma is a benign odontogenic tumour, and the patient always presents at a later stage when the tumour is already in an aggressive state. The finding of high mutation of *BRAF* V600E indicates the need to explore the molecular pathogenesis of ameloblastoma. However, there is inconsistent evidence regarding this mutation occurrence and its association with clinical information. This systematic review and meta-analysis aim to pool the overall mutation prevalence of *BRAF* V600E in reported ameloblastoma cases and to determine its association with patient demographic and clinicopathological features. This meta-analysis shows that *BRAF* V600E mutation has a high pooled prevalence of 70.49% in ameloblastoma. Furthermore, there was a significant meta-analysis association for those younger than 54 years old and in the mandible. Researchers could utilise these findings to improve the treatment option and find a possible new biomarker for the early detection of ameloblastoma.

**Abstract:**

The discovery that ameloblastoma has a high mutation incidence of *BRAF* V600E may enable a better investigation of pathophysiology. However, there is inconsistent evidence regarding this mutation occurrence and its association with clinical information. This systematic review and meta-analysis aim to pool the overall mutation prevalence of *BRAF* V600E in reported ameloblastoma cases and to determine its association with patient demographic and clinicopathological features. Following the PRISMA guidelines, a comprehensive article search was conducted through four databases (Scopus, Google Scholar, PubMed, and Web of Science). Seventeen articles between 2014 and 2022 met the inclusion criteria with 833 ameloblastoma cases. For each included study, the significance of *BRAF* V600E on the outcome parameters was determined using odd ratios and 95% confidence intervals. Meta-analysis prevalence of *BRAF* V600E in ameloblastoma was 70.49%, and a significant meta-analysis association was reported for those younger than 54 years old and in the mandible. On the contrary, other factors, such as sex, histological variants, and recurrence, were insignificant. As a result of the significant outcome of *BRAF* V600E mutation in ameloblastoma pathogenesis, targeted therapy formulation can be developed with this handful of evidence.

## 1. Introduction

Ameloblastoma is a benign, slow-growing epithelial odontogenic tumour. It is the second most common, constituting about 10% of all jaw neoplasms, and the annual pooled incidence rate of ameloblastoma was 0.92 cases per million [1,2,3]. Ameloblastoma affects both the maxilla and mandible. Due to the slow-growing nature of the tumour, it is usually neglected unknowingly at the early stage [1,4]. At a later stage, patients present with significant swelling and other accompanying signs and symptoms such as facial asymmetry, dental malocclusion, pain, and paraesthesia. In exceptional cases, it metastasises despite having a benign histologic appearance [4,5]. The mainstay treatment for ameloblastoma relies on surgical treatment; nonetheless, conservative treatments such as enucleation or curettage risk potential recurrence, whereas extensive surgical resection for massive ameloblastoma results in high morbidity and postoperative deformity [1,4].

For the past decade, the pathogenesis underlying ameloblastoma has unfolded. Ameloblastoma can be defined by uncontrolled cell proliferation, primarily driven by the mitogen-activated protein kinase (MAPK) signalling pathway, one of the main molecular pathways [5,6,7]. In this pathway, the most significant molecular event is the mutated *BRAF* gene, resulting in the substitution of amino acid valine (V) by glutamic acid (E) at position 600 (mutated *BRAF* V600E). Mutated *BRAF* V600E in this MAPK pathway enables the cells to proliferate excessively, leading to neoplasm formation [8]. It has been shown that this mutated *BRAF* V600E is commonly found in ameloblastoma of the mandible [9,10,11]. In contrast, for ameloblastoma developed in the maxilla, mutations of the protein Smoothened (SMO) of the Hedgehog pathway, a non-MAPK pathway, is involved [5].

Given the evolving molecular discoveries in ameloblastoma, this work presented a systematic review and meta-analysis to pool the mutation prevalence of *BRAF* V600E and to seek any association between *BRAF* V600E mutation and demographic profiles (age and sex) as well as clinicopathological features (site, histological variants, and recurrence) in ameloblastoma.

## 2. Materials and Methods

### 2.1. Research Questions

In this study, the following research questions were formulated: (1) What is the role of *BRAF* V600E mutation in ameloblastoma regarding its pooled prevalence, and (2) how does *BRAF* V600E mutation in ameloblastoma associate with sociodemographic profiles and clinicopathological features?

### 2.2. Protocol and Eligibility Criteria

The report presentation followed preferred Reporting Items for Systematic Reviews and Meta-Analyses (PRISMA) guidelines from the screening protocol to the final analysis [12]. The protocol has been registered in the PROSPERO database (CRD42022328296).

The inclusion criteria for the studies to be considered in this systematic review and meta-analysis were as follows: (1) studies related to *BRAF* V600E mutation in ameloblastoma; (2) the studies with adequate clinical information on at least three of the following features: age, sex, site, histological variants, and recurrence; and (3) English-language articles.

The exclusion criteria were as follows: (1) the studies reported as review papers, books, practice guidelines, letters, editorials, commentaries, case reports, and pilot studies; (2) articles on metastatic ameloblastoma and ameloblastic carcinomas; and (3) case studies with less than 10 patients.

### 2.3. Information Sources and Search Criteria

The search was conducted in Google Scholar, PubMed, Web of Science, and Scopus databases. The main fraction keywords according to the PICO tool of the article [13] were selected as follows: ameloblastoma (Population), *BRAF* (Indicator), clinicopathological features (Comparison), and recurrence (Outcome). The search strategy involved combinations of keyword concepts by medical subject heading (MeSH) terminology. The article search was done by 30 April 2022 using the keywords of ‘Ameloblastoma’, ‘B-Raf protein’, ‘Proto-Oncogene Protein B-Raf’, ‘*BRAF*’, and ‘*BRAF* V600E’.

### 2.4. Study Selection

Preliminarily, the selected articles were screened for validity and relevance with the inclusion and exclusion criteria regarding the information, selection bias, and quality of data analysis. Then, we proceeded with the article’s title and abstract reading to verify the content. Next, the screening process was done by reading the full-text articles to finalize which articles were eligible based on the study’s aims. The articles that did not fulfil the criteria and were out of scope were removed for each step. The final included articles have proceeded with the risk of bias assessment and quantitative analysis.

### 2.5. Data Collection Process and Data Items

Data were extracted by two authors independently (M.N.M.@Y. and N.R.A.R), and the third author, E.S.C., participated if any discrepancy was raised for analysis starting from the initial screening till the assessment of the bias. Relevant information was listed in a table as follows: authors and year of publication, demographic aspects of sex and age, number of cases, percentage of *BRAF* V600E positive mutation, tumour location, histological variants, and number of recurrences. Eligible, open-access or restricted-access articles were retrieved by Universiti Sains Malaysia (USM) library support. Research papers from those sources were uploaded into Mendeley reference manager software, and duplicate articles were removed.

### 2.6. Assessment of Risk of Bias in Individual Studies

The risks of bias in the selected studies were assessed using the Agency for Healthcare Research and Quality (AHRQ) modified scale for observational studies [14,15]. This scale assessment tool consists of nine main evaluation components with sub-elements. The evaluation was assessed for each study to obtain the overall score, which is a score for each component, with ‘adequate’ (A) when the criteria were fulfilled, ‘inadequate’ (I) when the criteria were not fulfilled ‘not reported’ (N) when the study failed to provide the required information, and ‘no information’ (-) when the criteria do not apply to the study design [15].

### 2.7. Statistical Methods 

The mutation pooled prevalence of *BRAF* V600E among ameloblastoma patients was analysed using Stata software (version 17, College Station, TX, USA) with a 95% confidence interval (CI) [16]. Cochrane Q of heterogeneity test is significant when the I-squared (I^2^) statistic value is more than 50% with the *p*-value less than 0.05. The value of meta-analysis was used depending on the fixed effect models (FEM) if the heterogeneity was not significant, and random or the quality effect models (QEM) were used if heterogeneity was significant [17].

A funnel plot of included studies was extracted from Stata software (version 17) and used to evaluate the risk of publication bias. Plotted graph with symmetrical distribution of inverted funnel shape and without outliers indicates a low risk of bias.

A meta-analysis of associations between *BRAF* V600E mutation in ameloblastoma and clinicopathological features were analysed using Review Manager software (RevMan version 5.4, London, UK). Heterogeneity test data from RevMan version 5.4 was evaluated depending on the chi-square (χ^2^) test and determined by the I^2^ statistic and statistical significance with a *p*-value of less than 0.05. Each association study was presented in forest plots to see the outcomes of individual studies’ effects and to conclude with overall pooled studies.

Age was divided into three age groups (young, adult, and older). First, age groups were determined by calculating the area under the curve of normal distribution using the IBM SPSS version 24. Then, the area under the curve was divided into four quarters (Q1, Q2, Q3, and Q4). The cut-off point for the young age group was Q1 and below, the older age group was Q4 and above, and the adult age group was a combination of Q2 and Q3.

## 3. Results

### 3.1. Search Sequence and Quality Assessment of Selected Publications

In total, 782 abstracts and titles were obtained through electronic database searches, and 175 articles were excluded due to duplicate articles (Figure 1). Then, the remaining 697 articles proceeded with screening by reading the titles and abstracts. Subsequently, 521 articles were excluded as they did not fulfil the criteria. Lastly, the relevance of 86 full-text articles was screened in detail. A total of 69 articles were excluded in the final step as did not meet the study aims, and reasons for excluded full-text articles were listed (Table 1). The remaining 17 studies were all evaluated for risk of bias according to AHRQ (Table 2). Elements which did not apply to the study design (-) were excluded from the domain summary. Only the minimum 50% of the elements in each domain were accounted for as an A score. Six studies were evaluated with an A score for nine domains [6,9,18,19,20,21], six studies with an A score for eight out of nine domains [22,23,24,25,26,27], three studies with an A score for seven out of nine domains [5,11,28], and two studies with an A score for six out of nine domains [29,30].

### 3.2. Study Characteristics

The final 17 studies were included for qualitative and quantitative analysis with a total of 833 patients. The research by Sweeney et al. in 2014 was the earliest study, and the latest publication was in 2022, by Kunmongkolwut and colleagues [26]. The most extensive study was by da Silva Marcelino et al. [25] and included 128 patients; the lowest number of samples was by Yukimori et al. [30], with 14 patients. A summary of the 17 selected studies and association variables with *BRAF* V600E mutation in ameloblastoma are summarised in Table 3.

### 3.3. Quantitative Synthesis 

#### 3.3.1. Prevalence of *BRAF* V600E Mutation

The total number of *BRAF* mutations for further analysis includes the number of positive mutations detected by polymerase chain reaction (PCR) [5,9,11,19,21,22,24,29,30], and data from immunohistochemistry (IHC) results whereby IHC was the sole method available in seven individual studies [18,20,23,25,26,27,28] (Table 3). For Brown et al.’s [6] study, IHC data were used because not all cases had PCR data; moreover, for cases with both IHC and PCR data, 100% concordance was recorded for both methods in this individual study.

Heterogeneity was significant for the pooled prevalence of *BRAF* mutation among ameloblastoma, which had a *p* < 0.05 in Cochrane Q statistics, and I^2^ statistics values of 83.09%. From 17 studies that reported total *BRAF* mutation cases, the overall pooled prevalence among ameloblastoma based on QEM was 70.49% (95% CI = 62.20–78.19%; *p* < 0.05) (Figure 2). Publication bias of this pooled prevalence was also evaluated using a funnel plot, showing a symmetrical plot, indicating a low potential risk of publication bias (Figure 3).

#### 3.3.2. *BRAF* V600E Mutation and Demographic Profiles

##### Age with *BRAF* V600E Mutation

Based on available data, a sensitivity study for the association between age and *BRAF* V600E mutation was conducted for 10 out of 17 studies. First, histogram and normal age distribution for total cases of ameloblastoma were plotted. From the quartile analysis, the age was then grouped into three: young (less and equal to 24 years old), adult (more than 24 years old and less than 54 years old), and older (more and equal to 54 years old) (Appendix A). Finally, the intergroup comparison was made of young versus adult, young versus older, and adult versus older.

For young versus adult comparison, young patients recorded 75.86% (66 out of 87 cases had a mutation of *BRAF* V600E), and adult patients recorded 79.01% (128 out of 162 cases had a mutation of *BRAF* V600E). FEM was used as there was no significant amount of heterogeneity (*p* = 0.82; I^2^ = 0%). The pooled analysis showed no significant association of *BRAF* V600E mutation in the young age group compared to the adult age group (OR = 0.95; 95% CI = 0.48–1.90; *p* = 0.89) (Figure 4A).

For the young versus older comparison, young patients recorded 75.86%, and older patients recorded 50.00% (47 out of 94 cases had a mutation of *BRAF* V600E). FEM was used as there was no significant amount of heterogeneity (*p* = 0.07; I^2^ = 45%). The pooled analysis showed a significant association of *BRAF* V600E mutation in the young age group compared to the older age group (OR = 2.87; 95% CI = 1.44–5.71; *p* = 0.003) (Figure 4B).

For adult versus older comparison, 79.01% of adult patients recorded a mutation of *BRAF* V600E, and 50.00% of older patients. FEM was used as there was no significant heterogeneity (*p* = 0.37; I^2^ = 8%). The pooled analysis showed a significant association of *BRAF* V600E mutation in the adult age group compared with the older age group (OR = 3.44; 95% CI = 1.83–6.48; *p* = 0.0001) (Figure 4C).

There was no significant association between *BRAF* V600E mutation for the young and adult groups. However, it was significant in the young age group compared with the older age group and the adult age group compared with the older age group. Therefore, *BRAF* V600E mutation was significantly associated with age less than 54 years old among ameloblastoma patients, as shown in the pooled analysis (OR = 3.42; 95% CI = 1.94–6.04; *p* < 0.0001) based on FEM, as the heterogeneity was not significant (*p* = 0.11; I^2^ = 38%). (Figure 4D).

##### Sex with *BRAF* V600E Mutation

The association analysis between *BRAF* V600E mutation and sex proceeded with 17 studies of 590 patients. Among 465 male patients, 70.54% of patients were *BRAF* positive. Females showed a slightly higher percentage than males (68.63%), consisting of 245 out of 357 female patients. FEM data used as the heterogeneity test was not significant (*p* = 0.31; I^2^ = 12%). There was no association between *BRAF* V600E mutation and sex, as the statistical analysis was not significant (OR = 1.14; 95% CI = 0.83–1.57: *p* = 0.41) (Figure 5A).

#### 3.3.3. *BRAF* V600E Mutation and Clinicopathological Features Association

##### Tumour Location with *BRAF* V600E Mutation

A sensitivity study was conducted for an association between tumour location (mandible versus maxilla) and *BRAF* V600E mutation for 12 out of 17 studies based on available data. 74.67% of the mandible (336 out of 450 patients) and 30.00% of the maxilla (30 out of 100 patients) had *BRAF* V600E mutation. QEM data used as the heterogeneity test was significant (*p* = 0.01; I^2^ = 53%). There was an association between the mandible and *BRAF* V600E mutation, as the statistical analysis proved significant (OR = 5.24; 95% CI = 1.96–13.98; *p* = 0.0009) (Figure 5B).

##### Recurrence with *BRAF* V600E Mutation

Based on available data, a sensitivity study for the association between recurrence and *BRAF* V600E mutation was conducted for 15 out of 17 studies. Of 175 recurrence cases, 120 (68.57%) were *BRAF* V600E positive. First presentation or primary cases reported mutations in 400 out of 560 patients (71.43%). FEM data used as the heterogeneity test was not significant (*p* = 0.02; I^2^ = 48%). There was no association between recurrence and *BRAF* V600E mutation (OR = 0.92; 95% CI = 0.62–1.1.37; *p* = 0.69) (Figure 5C).

##### Histological Variants with *BRAF* V600E Mutation

A sensitivity study for the association between histological variants and *BRAF* V600E mutation was conducted for 11 of 17 studies based on available data. Out of 432 conventional ameloblastoma cases, 316 (73.15%) had *BRAF* mutation, while other variants (including unicystic, desmoplastic, and peripheral) reported that 80.33% (98 out of 122 patients) also had the mutation. FEM was used as there was no significant amount of heterogeneity (*p* = 0.30; I^2^ = 10%). There was no significant association between histological variants and *BRAF* V600E mutation (OR = 0.72; 95% CI = 0.43–1.20; *p* = 0.20) (Figure 6A).

Other histological variants were also reported with the same range of *BRAF* V600E mutation with conventional ameloblastoma: 82.11% mutation in unicystic, 62.50% in desmoplastic, and 66.67% mutation in peripheral ameloblastoma. Further subgroup analysis of histological variants of conventional with unicystic, desmoplastic, and peripheral ameloblastoma also showed no significant association with *BRAF* V600E mutation (Figure 6B–D).

## 4. Discussion

Ameloblastoma is an aggressive benign odontogenic tumour of the jaws. Brown et al. [6] have found the mutation in the MAPK pathways, remarkably high in the *BRAF* gene. More researchers then began to explore and refine this discovery of oncogenic mutation, correlating it to the clinical implication for the improvement in disease control and treatments [29]. This systematic review and meta-analysis were conducted to identify the pooled prevalence of *BRAF* mutation and associate it with the sociodemographic and clinicopathological features.

This study reported a 70.49 % mutation prevalence with the *BRAF* V600E gene among ameloblastoma patients. Among individual studies, the lowest prevalence of *BRAF* V600E mutation was reported by Shirsat et al. [28] (33.33%), and the highest prevalence (92.00%) was reported by Derakhshan et al. [11]. This meta-analysis outcome of high prevalence has shown that *BRAF* V600E mutation has played a role in molecular pathogenesis in ameloblastoma incidence. Even though the status and exact mechanisms are still unclear, a high prevalence status may indicate a new treatment area exploration concerning this specific mutation. Previously, the *BRAF* V600E mutation focused on malignant tumours, where the prevalence was approximately 74.6% for papillary thyroid carcinoma [95], 7.4% for colorectal cancer [96], and 60% for melanomas [97]. According to previous studies, several gene mutations have been identified in the background of *BRAF* V600E positive mutation in ameloblastoma, including somatic mutation in cyclin-dependent kinase inhibitor 2A (*CDKN2A*), catenin beta 1 (*CTNNB1*), fibroblast growth factor receptors (*FGFR*), Kirsten rat sarcoma virus (*KRAS*), phosphatidylinositol-4,5-bisphosphate 3-kinase catalytic subunit alpha (*PIK3CA*), and phosphatase and tensin homolog (*PTEN*) [19,98].

There was also a case report which was ameloblastoma initially but later presented with ameloblastic carcinoma; positive *BRAF* V600E mutation was subsequently detected [99]. This malignant counterpart also expresses *BRAF* V600E mutation (about 25 to 33%) [100,101]. Hence, there is a possibility that *BRAF* V600E mutation may play a role in the malignant transformation of ameloblastoma, as this rare odontogenic malignancy has close features that combine the histologic features of ameloblastoma with cytologic atypia [100,101,102].

Having a mutation of *BRAF* V600E will alter the MAPK pathway, and Brown et al. [7] suggested this alteration may be crucial in the early stage of ameloblastoma pathogenesis. Other than the *BRAF* gene in the MAPK pathway, *FGFR2* and *RAS* mutations were also part of the pathogenesis of most ameloblastoma cases [6]. On the other hand, the Hedgehog pathway has also been reported as the secondary mutation, specifically the somatic mutation of Smoothened (*SMO*) gene [5,7].

The *BRAF* V600E mutation remains the most critical molecular marker studied in ameloblastoma pathogenesis [9,33,37,56]. Hence, a valid method for detecting this mutation is crucial in accurately evaluating mutation occurrence, either by IHC or molecular assay, usually by PCR. PCR is the gold standard for detecting any gene mutation. However, due to its costliness and the limited number of facilities that offers this technique, *BRAF* V600E mutation was rarely assessed. Later, establishing a mouse monoclonal antibody (VE-1) allowed for the analysis of IHC for the *BRAF* V600E mutation in numerous tumours [103,104]. Therefore, IHC is a preferred technique as it is easy, reliable, affordable, and widely used as a standard diagnostic histopathological procedure [55,72].

These two methods reported some discrepancies, mainly false-positive and false-negative results. For example, colorectal carcinomas [105] and melanomas [9,106] had false-positive results from the IHC assessment but were negative in the molecular assay. Even in this meta-analysis, some studies reported a few differences in the *BRAF* V600E mutation detection by IHC compared with PCR [9,11,21]. In those mentioned studies, when there were false-positive or false-negative cases, further molecular assays such as DNA sequencing were performed to confirm the mutation status [9,11,21].

IHC false positives are attributed to sampling contamination with other tissues with positive immunoreactivity [104,107]. In addition, IHC false negatives are possible due to decalcified old histological samples. Therefore, it is suggested that tissues be preserved and processed thoroughly within two hours following collection [103]. Other likely explanations of IHC false negatives include loss of expression of the altered antigen, such as in necrotic tumour regions, and additional mutations that prevent the mutated messenger RNA from being translocated into a functional protein [104,108].

Although they possess some pitfalls, several studies have proven a strong concordance between IHC and molecular assays [103,109,110]. Thus, IHC staining for *BRAF* V600E detection is a reliable technique compared to PCR for identifying the *BRAF* V600E mutation in various tumours, and it has the benefit of substantial cost and labour savings. In addition, the identification of *BRAF* V600E mutation using the IHC method should be interpreted carefully by expert pathologists [72].

This meta-analysis reported that the percentage of *BRAF* V600E mutation is higher in patients less than 54 years old—the age group for the young and adults. However, most individual studies reported no association between mutation status and age [9,29,91]. Thus, an objective comparison was made in this study by plotting a normal distribution graph of total ameloblastoma cases in all eligible studies. As a result, the meta-analysis revealed a significant association between *BRAF* V600E mutation and young and adult age groups among ameloblastoma patients.

The association between age and disease risk has always been one of the primary criteria for epidemiology [111,112]. Our findings showed that the younger generation (less than 54 years old) was associated with *BRAF* V600E mutation in ameloblastoma patients. Based on this result, we suggest a correlation between odontogenesis and *BRAF* V600E mutation, which led to the ameloblastoma incidence. Fibroblast growth factor (FGF) is one of the signalling molecules in mammalian tooth development, which initiates signalling through multiple downstream intracellular pathways and later activates Ras signals, including RAF/MEK/ERK [113]. The *BRAF* V600E mutation essentially activates MEK/ERK signalling, leading to tumour formation [8] and, in this case, ameloblastoma formation. This finding may reflect why the young age group has a significant association, as odontogenesis is more prevalent. In resource-limited settings, *BRAF* V600E mutation screening should be prioritised for patients below the age of 54, as proven by our findings. Besides, they also have a longer lifespan and thus have a higher risk of recurrence. Testing in advance will offer an option for better treatment with targeted therapies against *BRAF* V600E mutation that could prevent the tumour from recurring.

Sex can sometimes significantly influence disease formation [114]. Hence, health professionals investigate the influence of sex in many ways in terms of aetiology, diagnosis, progression, prevention, treatment, health outcomes of disease, and exposure to risk [114]. In the current study, the statistical analysis failed to prove association between sex and mutation occurrence. This finding supported previous research showing that mutation of *BRAF* V600E is not affected by sex in ameloblastoma patients.

The studies of tumour site-specific mutations in ameloblastoma were first reported in 2014 and were proposed as a new paradigm [5,6]. Many studies have reported a higher frequency of association between *BRAF* V600E mutation and the mandible and showed a statistically significant result compared with the maxilla [5,23,32]. The result is in line with the current report. On the other hand, five studies revealed no significant differences between the mandible or maxilla tumour locations [11,18,22,29,30]. The effects of sample sizes can explain the differences in the results. A meta-analysis helps pool the included studies; hence, a better sample size calculation can be generated and more representative. According to Sweeney et al. [5], the SMO gene mutation substantially affects the maxilla rather than the mandible, which may explain the independent odontogenic pathways in the jaw location [115]. On a broader spectrum, this finding emphasised the understanding of the anatomical specificity in mutation-driven pathogenesis, reflecting the distinctive developmental pathways of the jaws [5]. Significant mutation of *BRAF* V600E in mandibular ameloblastoma may allow for better risk assessment and the possibility of personalized adjunctive therapy to sustain jaw functionality.

Furthermore, there was no significant difference between the mutation of *BRAF* V600E and ameloblastoma histological variants via meta-analysis. Only a study by Gültekin et al. [19] showed a significant association with conventional ameloblastoma. It investigates whether *BRAF* mutation occurs in different variants at a similar proportion [29]. They concluded that all histological variants of ameloblastoma underwent similar molecular alterations for this benign odontogenic neoplasm [29].

Our meta-analysis revealed that the mutation of *BRAF* V600E was not significantly associated either with the first presentation or the recurrence cases. It was consistent with most studies, except one study had an association of *BRAF* V600E positivity with the recurrence cases with an odds ratio of 11.45 [18]. On the other hand, *BRAF* wild type was found to have an earlier recurrence, especially in those treated with surgical enucleation rather than surgical resection [2,116]. Maxilla has a higher recurrence rate, most probably due to the anatomy, causing limited treatment options and difficulty achieving a safe and clear surgical margin [11,116]. However, current studies on the relationship between *BRAF* V600E and the recurrences are still unclear. Further studies that provide a definite conclusion on this matter can improve the clinical management of ameloblastoma [2,117].

*BRAF* V600E mutation has a significant association with a broad range of neoplasms [105,106,118,119], and in ameloblastoma specifically [5,6,18,19,21,22]. Furthermore, once the mutation occurs, the complicated MAPK pathway, which involves the activation of downstream RAS, RAF, MEK, and ERK, becomes activated in tumorigenesis [120]. Hence, each MAPK component that undergoes mutation needs to be understood to formulate the best treatment regimen for ameloblastoma patients [5,6,7,37].

Due to its benign and locally aggressive behaviour, current ameloblastoma management is by surgical intervention. However, these standard surgical treatments are either by resection or conservative (curettage or enucleation), with the latter possessing a higher recurrence rate [121]. The meta-analysis result has significantly upheld the correlation of the *BRAF* gene to ameloblastoma occurrence. Hence, taking advantage of this molecular pathway, a *BRAF* inhibitor treatment may be used to avoid wide surgical resection or multiple surgical procedures due to recurrence [7].

*BRAF* inhibitor studies actively explored their effectiveness, where the data collected shows promising results. A study on ameloblastoma cell lines has reported an in vitro sensitivity of *BRAF* inhibitor (vemurafenib) to hinder V600E mutation [5,6]. A case report of a 29-year-old woman with a recurrence of ameloblastoma with *BRAF* V600E mutation, who received vemurafenib, was symptomless and had tumour shrinkage after 11 months of therapy [62]. Vemurafenib is a well-known *BRAF* inhibitor that the FDA has authorised for treating metastatic melanoma with *BRAF* V600E [122]. Another inhibitor, dabrafenib, is also used to control the MAPK pathway with different neoplasms, including ameloblastoma [123,124]. Faden and Algazi [36], and Tan et al. [67] have reported using dabrafenib in recurrence cases of ameloblastoma, which also showed promising results. This limited clinical data reflected that this inhibitor therapy seems to be an effective treatment modality. However, there were downsides to this therapy, such as the development of resistance and acquiring skin tumours, thus proposing a dual-agent therapy instead of single-agent therapy [56].

The current study profiled the relationship between the mutation of *BRAF* V600E and the incidence of ameloblastoma. We have found significant findings related to the age groups and tumour location. However, some limitations might influence the interpretation. Firstly, we could not do the meta-analysis on each method (IHC versus molecular assay) in detecting *BRAF* V600E due to a lack of data from the search strategy criteria. Therefore, we suggest that future research on a meta-analysis related to comparing the validated methods to detect *BRAF* V600E mutation in ameloblastoma. For example, this has been done in the study of papillary thyroid carcinoma, a type of cancer with significant *BRAF* V600E mutation [125]. Secondly, the definition of recurrences in the studies was not clear. Most studies reported the cases as primary or recurrence cases without specifying whether it was a true recurrence or a residual tumour. Therefore, despite no significant finding in the current study, the chance of *BRAF* V600E mutation occurring in recurrence cases versus primary presentation is still debatable. Thus, it is recommended that clinicians and researchers record those details when documenting a case report, which is essential for further analysis and later can be translated into clinical management. Finally, data on treatment options for ameloblastoma has been excluded in this review due to the limited information available in the articles. It may be because *BRAF* inhibitor is not widely used in treating ameloblastoma. Therefore, further clinical trials are recommended to determine the effectiveness of this drug in ameloblastoma management. Once the data is more widely available, future studies on this aspect should be explored by meta-analysis. This may help improve the clinical outcome of ameloblastoma patients.

## 5. Conclusions

The present systematic review and meta-analysis show that *BRAF* V600E mutation has a high pooled prevalence of 70.49% in ameloblastoma. Furthermore, a significant meta-analysis association was reported for those younger than 54 years old, and in the mandible. On the contrary, other factors, such as sex, histological variants, and recurrence, were insignificant among ameloblastoma cases with *BRAF* V600E mutation. Researchers could utilise these findings to improve the treatment option and find a possible new biomarker for the early detection of ameloblastoma. This evidence-based medicine information is essential in targeted therapy development. However, further well-designed cohort studies are needed to verify the association of *BRAF* V600E mutation in ameloblastoma before applying new medical interventions.

## Figures and Tables

**Figure 1 cancers-14-05593-f001:**
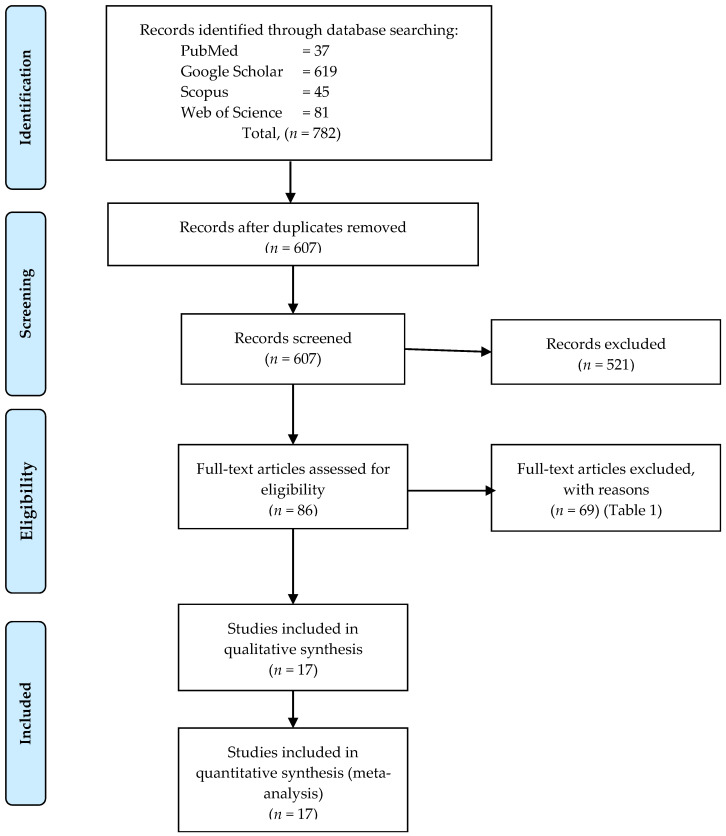
PRISMA flow diagram of study selection and screening [12].

**Figure 2 cancers-14-05593-f002:**
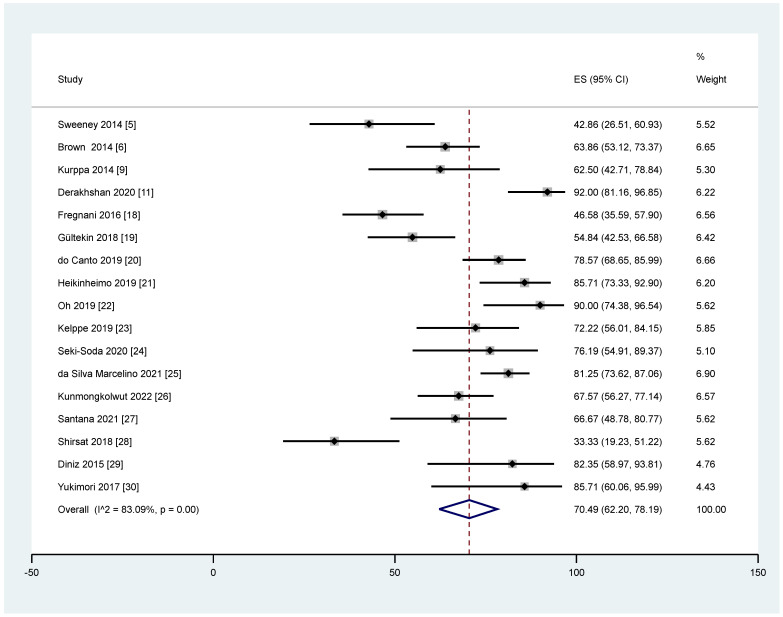
The pooled prevalence of *BRAF* V600E mutation in ameloblastoma cases.

**Figure 3 cancers-14-05593-f003:**
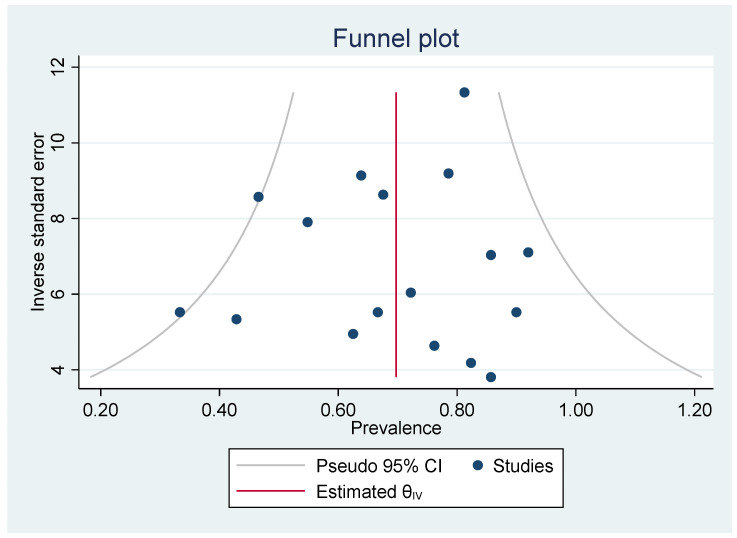
Funnel plot for publication bias evaluation.

**Figure 4 cancers-14-05593-f004:**
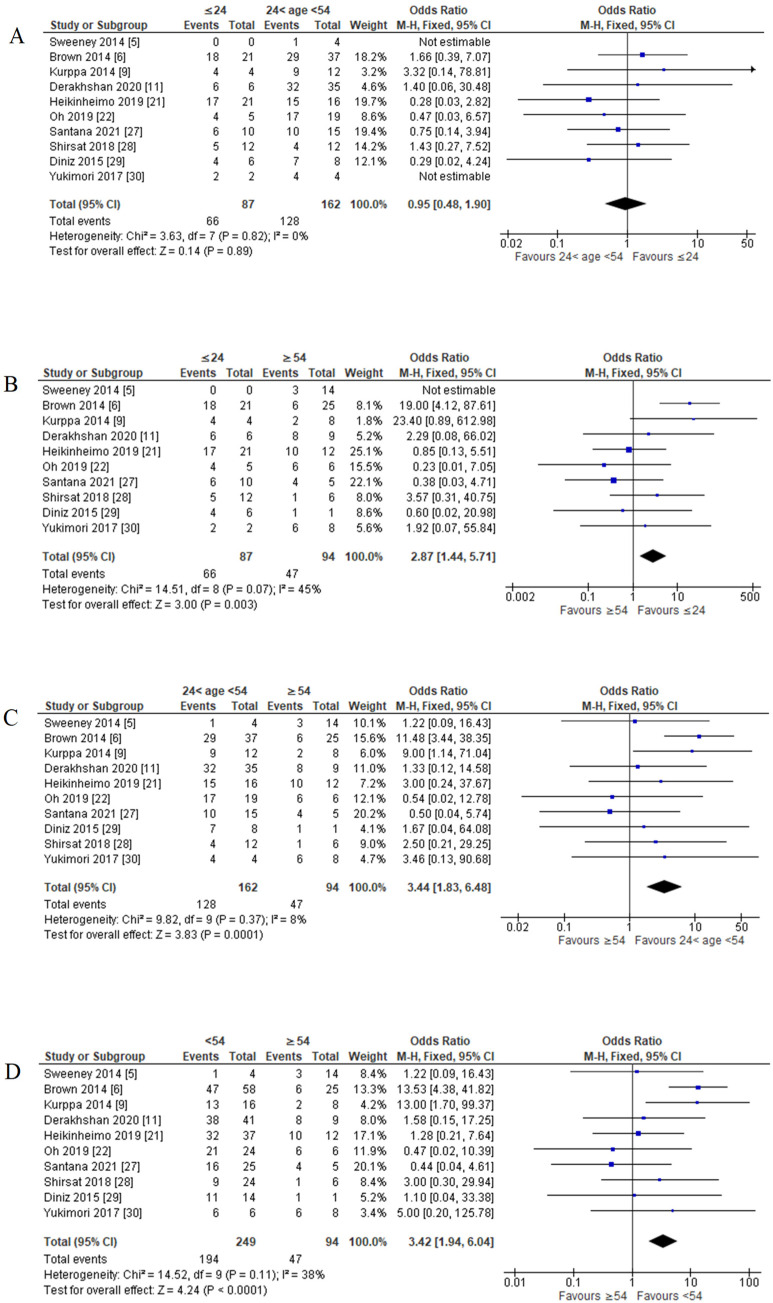
Forest plot of age groups with *BRAF* V600E mutation in ameloblastoma cases. (**A**) Young age group versus adult age group; (**B**) young age group versus older age group; (**C**) adult age group versus older age group; (**D**) young and adult age group versus older age group.

**Figure 5 cancers-14-05593-f005:**
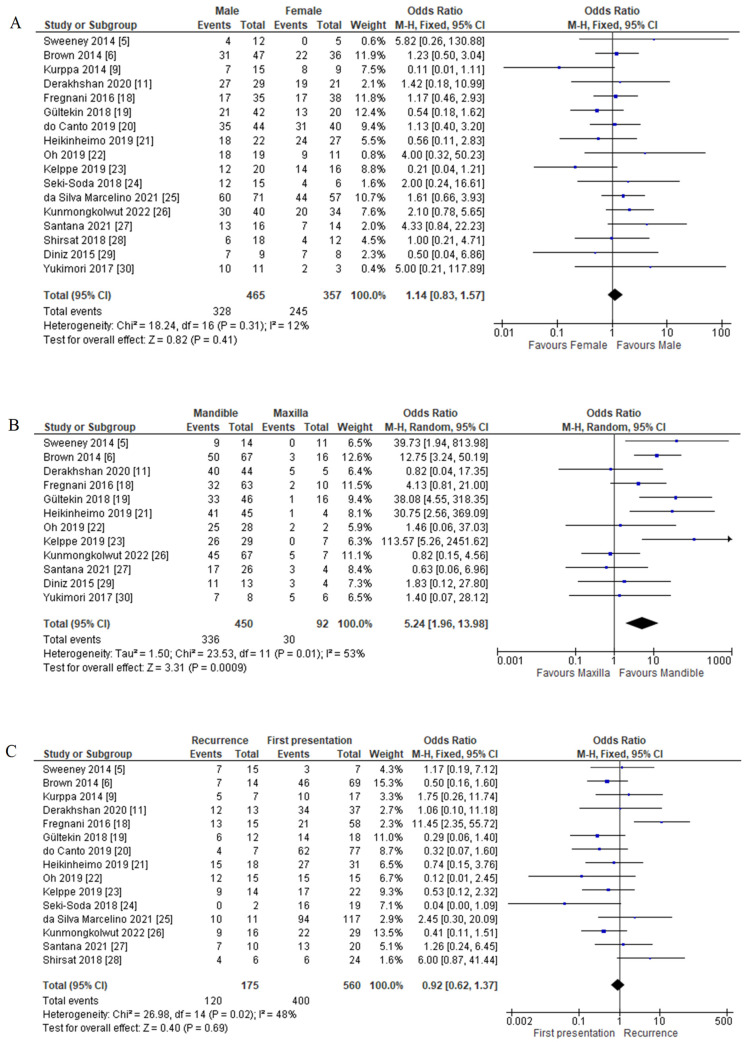
Forest plot of clinicopathological features association with *BRAF* V600E mutation in ameloblastoma cases. (**A**) Sex with *BRAF* V600E mutation; (**B**) tumour location with *BRAF* V600E mutation; (**C**) recurrence with *BRAF* V600E mutation.

**Figure 6 cancers-14-05593-f006:**
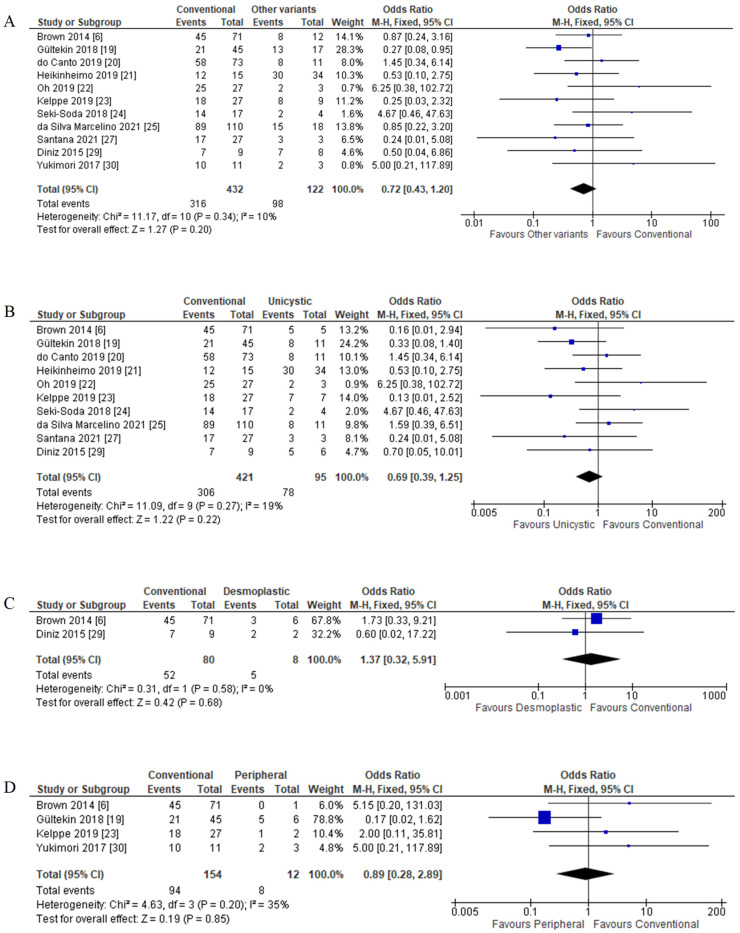
Forest plot of histological variants association with *BRAF* V600E mutation in ameloblastoma cases. (**A**) Conventional versus other variants; (**B**) conventional versus unicystic; (**C**) conventional versus desmoplastic; (**D**) conventional versus peripheral.

**Table 1 cancers-14-05593-t001:** Full-text articles excluded with reason.

Articles Excluded	Reason for Exclusion	No. of Articles
Abe et al., 2018 [31]	Letter/correspondence/commentary/ response/communication	13
Abe et al., 2018 [32]		
Brunnet et al., 2015 [33]		
De Sousa et al., 2016 [34]		
Coura et al., 2021 [35]		
Faden and Algazi, 2017 [36]		
Gomes et al., 2014 [37]		
Kaye et al., 2015 [38]		
Kaye et al., 2017 [39]		
Magliocca et al., 2016 [40]		
Mota santana et al., 2020 [41]		
Saffari et al., 2019 [42]		
Waqa et al., 2020 [43]		
Effiom et al., 2018 [2]	Review paper	17
McClary et al., 2016 [4]		
Brown and Betz, 2015 [7]		
Diniz et al., 2017 [44]		
Daws et al., 2021 [45]		
do Canto et al., 2016 [46]		
Fuchigami et al., 2021 [47]		
Heikinheimo et al., 2015 [48]		
Jhamb and Kramer, 2014 [49]		
Khalele and Al-Shiaty, 2016 [50]		
Kreppel and Zöller, 2018 [51]		
Marín et al., 2021 [52]		
Martins-de-Barros et al., 2022 [53]		
Ngan et al., 2022 [54]		
Ritterhouse and Barletta, 2015 [55]		
Shi et al., 2021 [56]		
You et al., 2019 [57]		
Abramson et al., 2022 [58]	Case report	10
Bernaola-Paredes et al., 2021 [59]		
Broudic-Guibert et al., 2019 [60]		
Brunet et al., 2019 [61]		
Fernandes et al., 2018 [62]		
Hirschhorn et al., 2021 [63]		
Roque and Yazmin, 2017 [64]		
Rotellini et al., 2016 [65]		
Suzuki et al., 2020 [66]		
Tan et al., 2016 [67]		
Bartels et al., 2018 [68]	Number of cases less than 10	9
Diniz et al., 2017 [69]		
Kennedy et al., 2016 [70]		
Kondo et al., 2020 [71]		
Pereira et al., 2016 [72]		
Sant’Ana et al., 2021 [73]		
Shi et al., 2021 [74]		
Shimura et al., 2020 [75]		
You et al., 2019 [76]		
Bologna-Molina et al., 2019 [77]	Unrelated to *BRAF* studies, failure to provide clinical information or inability to present data clearly	17
Bonacina et al., 2022 [78]		
Coura et al., 2020 [79]		
Duarte-Andrade et al., 2019 [80]		
Fujii et al., 2022 [81]		
Guan et al., 2019 [82]		
Kokubun et al., 2022 [83]		
Lapthanasupkul et al., 2021 [84]		
Oh et al., 2021 [85]		
Oh et al., 2022 [86]		
Owosho et al., 2021 [87]		
Peralta et al., 2019 [88]		
Salama et al., 2020 [89]		
Sharp et al., 2019 [90]		
Soltani et al., 2018 [91]		
Tseng et al., 2022 [92]		
Zhang et al., 2020 [93]		
Diniz et al., 2017 [94]	Samples were taken from the same Hospital as Diniz et al., 2015 [29]	1

**Table 2 cancers-14-05593-t002:** Risk of bias among studies analysed.

Domain	Elements	Sweeney 2014 [5]	Brown 2014 [6]	Kurppa 2014 [9]	Derakhshan 2020 [11]	Fregnani 2016 [18]	Gültekin 2018 [19]	do Canto 2019 [20]	Heikinheimo 2019 [21]	Oh 2019 [22]	Kelppe 2019 [23]	Seki-Soda 2020 [24]	da Silva Marcelino 2021 [25]	Kunmongkolwut 2022 [26]	Santana 2021 [27]	Shirsat 2018 [28]	Diniz 2015 [29]	Yukimori 2017 [30]
Study question	Clearly focused and appropriate question	A	A	A	A	A	A	A	A	A	A	A	A	A	A	A	A	A
Study population	Description of study population	A	A	A	A	A	A	A	A	A	A	A	A	A	A	A	A	A
	Sample size justification	A	A	A	A	A	A	A	A	A	A	A	A	A	A	A	A	A
Comparability of subjects	Specific inclusion/exclusion criteria	A	A	A	A	A	A	A	A	A	A	A	A	A	A	A	N	A
	Criteria applied equally to all groups	A	A	A	A	A	A	A	A	A	A	A	A	A	A	A	N	A
	Comparability of groups at baseline with regard to disease status and prognostic factors	_	_	_	_	_	_	_	_	_	_	_	_	_	_	_	_	_
	Study groups comparable to non-participants with regard to confounding factors	_	_	_	_	_	_	_	_	_	_	_	_	_	_	_	_	_
	Use of concurrent controls	_	_	_	_	_	_	_	_	_	_	_	_	_	_	_	_	_
	Comparability of follow-up among groups	_	_	_	_	_	_	_	_	_	_	_	_	_	_	_	_	_
Exposure or intervention	Clear definition of exposure	A	A	A	A	A	A	A	A	A	A	A	A	A	A	A	A	A
	Measurement method standard, valid, and reliable	A	A	A	A	A	A	A	A	A	A	A	A	A	A	A	A	A
	Exposure measured equally in all study groups	A	A	A	A	A	A	A	A	A	A	A	A	A	A	A	A	A
Outcome measurement	Primary/secondary outcomes clearly defined	A	A	A	A	A	A	A	A	A	A	A	A	A	A	A	A	A
	Outcomes assessed blind to exposure or intervention status	_	_	_	_	_	_	_	_	_	_	_	_	_	_	_	_	_
	Method of outcome assessment standard, valid, and reliable	A	A	A	A	A	A	A	A	A	A	A	A	A	A	A	A	A
	Length of follow-up adequate for question	_	_	_	_	_	_	_	_	_	_	_	_	_	_	_	_	_
Statistical analysis	Statistical tests appropriate	A	A	A	A	A	A	A	A	A	A	A	A	A	A	A	N	N
	Multiple comparisons taken into consideration	N	A	A	N	A	A	A	A	A	A	A	A	A	A	N	A	A
	Modelling and multivariate techniques appropriate	N	A	N	N	A	N	A	A	N	N	N	A	A	N	N	N	N
	Power calculation provided	N	A	A	N	A	A	I	A	A	A	A	A	A	A	N	N	N
	Assessment of confounding variables	_	_	_	_	_	_	_	_	_	_	_	_	_	_	_	_	_
	Dose–response assessment, if appropriate	_	_	_	_	_	_	_	_	_	_	_	_	_	_	_	_	_
Results	Measure of effect for outcomes and appropriate measure of precision	A	A	A	I	A	A	A	A	A	A	A	A	A	A	A	I	A
	Adequacy of follow-up for each study group	_	_	_	_	_	_	_	_	_	_	_	_	_	_	_	_	_
Discussion	Conclusions supported by results with biases and limitations taken into consideration	A	A	A	A	A	A	A	A	A	A	A	A	A	A	A	A	N
Funding or sponsorship	Type and sources of support for study	N	A	A	A	A	A	A	A	I	N	N	N	I	A	I	A	N

Abbreviations: A, adequate; I, inadequate; N, not reported; -, not applicable to the study design.

**Table 3 cancers-14-05593-t003:** The summary of 17 included ameloblastoma studies with *BRAF* V600E, demographic and clinicopathological features profile.

Author/ Year	Country	No. of Cases (*n*)	*BRAF*+, Detection Method (*n*)		No. of *BRAF*+, *n* (%) ^a^	Demographic (*BRAF*+/Total Case) (*n*)		Clinicopathological Features (*BRAF*+/Total Case) (*n*)		
			PCR	IHC		Sex	Age Group ^b^	Location	Histological Variant	Recurrence
Brown et al., 2014 [6]	USA	83	Pos = 30 Neg = 19 NA = 34	Pos = 53 Neg = 30	53(63.9)	M = 31/47 F = 22/36	Young = 18/21 Adult = 29/39 Older = 6/25	Mn = 50/67 Mx = 3/16	CA = 45/71 UA = 5/5 DA = 3/6 PA = 0/1	Yes = 7/14 No = 46/69
da Silva Marcelino et al., 2021 [25]	Brazil	128	NA	Pos = 104 Neg = 24	104(81.2)	M = 60/71 F = 44/57	NA	Mn = 104/128	CA = 89/110 UA = 15/18	Yes = 10/11 No = 94/117
Derakhshan et al., 2020 [11]	Iran	50	Pos = 46 Neg = 4	Pos = 39 Neg = 11	46(92.0)	M = 27/29 F = 19/21	Young = 6/6 Adult = 32/35 Older = 8/9	Mn = 40/44 Mx = 5/5 Both = 1/1	NA	Yes = 12/13 No = 34/37
Diniz et al., 2015 [29]	Brazil	17	Pos = 14 Neg = 3	NA	14(82.4)	M = 7/9 F = 7/8	Young = 4/6 Adult = 7/8 Older = 1/1 NA = 2	Mn = 11/13 Mx = ¾	CA = 7/9 UA = 5/6 DA = 2/2	NA
do Canto et al., 2016 [20]	Brazil	84	NA	Pos = 66 Neg = 18	66(78.6)	M = 35/44 F = 31/40	NA	Mn = 66/84	CA = 58/73 UA = 8/11	Yes = 4/7 No = 62/77
Fregnani et al., 2017 [18]	Brazil	73	NA	Pos = 34 Neg = 39	34(46.6)	M = 17/35 F = 17/38	NA	Mn = 32/63 Mx = 2/10	NA	Yes = 13/15 No = 21/58
Gültekin et al., 2018 [19]	Germany	62	Pos = 34 Neg = 28	NA	34(54.8)	M = 21/42 F = 13/20	NA	Mn = 33/46 Mx = 1/16	CA = 21/45 UA = 8/11 PA = 5/6	Yes = 6/12 No = 14/18 NA = 32
Heikinheimo et al., 2019 [21]	Finland	49	Pos = 42 Neg = 7 NA = 5	Pos = 39 Neg = 11 NA = 4	42(85.7)	M = 18/22 F = 24/27	Young = 17/21 Adult = 15/16 Older = 10/12	Mn = 41/45 Mx = 1/4	CA = 12/15 UA = 30/34	Yes = 15/18 No = 27/31
Kelppe et al., 2019 [23]	Finland	36	NA	Pos = 26 Neg = 10	26(72.2)	M = 12/20 F = 14/16	NA	Mn = 26/29 Mx = 0/7	CA = 18/27 UA = 7/7 PA = 1/2	Yes = 9/14 No = 17/22
Kunmongkol-wut et al., 2022 [26]	Thai	74	NA	Pos = 50 Neg = 24	50(67.6)	M = 30/40 F = 20/34	NA	Mn = 45/67 Mx = 5/7	NA	Yes = 9/16 No = 22/29 NA = 29
Kurppa et al., 2014 [9]	Finland	24	Pos = 15 Neg = 9	Pos = 11 Neg = 9 NA = 4	15(62.5)	M = 7/15 F = 8/9	Young = 4/4 Adult = 9/12 Older = 2/8	Mn = 15/24	CA = 15/24	Yes = 5/7 No = 10/17
Oh et al., 2019 [22]	Korea	30	Pos = 27 Neg = 3	Pos = 17 Neg = 10 NA = 3	27(90.0)	M = 18/19 F = 9/11	Young = 4/5 Adult = 17/19 Older = 6/6	Mn = 25/28 Mx = 2/2	CA = 25/27 UA = 2/3	Yes = 12/15 No = 15/15
Santana et al., 2021 [27]	Brazil	30	NA	Pos = 20 Neg = 10	20(66.7)	M = 13/16 F = 7/14	Young = 6/10 Adult = 10/15 Older = 4/5	Mn = 17/26 Mx = 3/4	CA = 17/27 UA = 3/3	Yes = 7/10 No = 13/20
Seki-Soda et al., 2020 [24]	Japan	21	Pos = 16 Neg = 5	Pos = 20 Neg = 1	16(76.2)	M = 12/15 F = 4/6	NA	Mn = 16/21	CA = 14/17 UA = 2/4	Yes = 0/2 No = 16/19
Shirsat et al., 2018 [28]	India	30	NA	Pos = 10 Neg = 20	10(33.3)	M = 6/18 F = 4/12	Young = 5/12 Adult = 4/12 Older = 1/6	Mn = 10/30	NA	Yes = 4/6 No = 6/24
Sweeney et al., 2014 [5]	USA	28	Pos = 12 Neg = 16	NA	12(42.9)	M = 4/13 F = 0/5 NA = 8/10	Young = 0/0 Adult = 1/4 Older = 3/14 NA = 8	Mn = 9/14 Mx = 0/11 Other = 3/3	NA	Yes = 7/15 No = 3/9 NA = 4
Yukimori et al., 2017 [30]	Japan	14	Pos = 12 Neg = 2	Pos = 12 Neg = 2	12(85.7)	M = 10/11 F = 2/3	Young = 2/2 Adult = 4/4 Older = 6/8	Mn = 7/8 Mx = 5/6	CA = 10/11 PA = 2/3	NA

Abbreviations: ^a^, total no. of *BRAF* mutations used for prevalence analysis; ^b^, age group, young (≤24), adult (24 < x < 54), older (≥54); NA, data not available/unknown status; Pos, positive; Neg, negative; M, male; F, female; Mn, mandibular; Mx, maxilla; CA, conventional ameloblastoma; UA, unicystic ameloblastoma; DA, desmoplastic ameloblastoma; PA, peripheral ameloblastoma.

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
