# Peer review of "BRAF V600E Mutation in Ameloblastoma: A Systematic Review and Meta-Analysis"

_cancers, 2022, doi:10.3390/cancers14225593_

Round 1

Reviewer 1 Report

It is a very comprehensive and well-written article that updates all existing information on the influence of the BRAFV600E mutation on ameloblastoma biology.

It seems to me that it would be much more complete if it mentioned exactly what is the percentage of cases with mutation reported in the ameloblastoma variants (not only those of the conventional ameloblastoma), I would like to know the data of the desmoplastic and peripheral ameloblastoma for example (it does not matter if they have similar percentages, it would be interesting to know the exact number of percentage) this data would be useful for reference in future works.

Although I know that it is not an objective of this study, it would also be good to report the existence of this mutation in ameloblastic carcinomas in the discussion.

It is important that the authors include ALL the selected full-text articles (all 86 articles) in their references, including those in Table 1 (Table 1. Full-text articles excluded with reason.) Readers must have the full data for future references.

Please update the terminology according to the 2022 WHO classification of odontogenic tumors.

Reviewer 2 Report

Dear authors,
Congratulations on the brilliant study that was well delineated and written. I have a few suggestions to improve the manuscript and make it actualized.

Firstly, I strongly suggest you substitute all "gender" for "sex" terms. After reading the WHO definition, and Nature's article about gender and sex differences, I am sure that it is necessary to make this differentiation in scientific biological studies. Therefore, I invite you to see the following articles and decide: https://www.who.int/health-topics/gender#tab=tab_1 and https://www.nature.com/articles/s41580-022-00467-w#Sec4.

The WHO updated the odontogenic tumor classification, and the "multicystic ameloblastoma" term is no longer used. Please, change this term to "conventional ameloblastoma" (explanation in PMID:  35578902).

Kind regards,

Reviewer 3 Report

Overall, the idea of the study is interesting. The manuscript is well-written, and the results could have merit in the related field. Just some concerns should be addressed.

 General comments:

- The authors should ensure that all gene names are italicized to match the standards of HUGO for gene nomenclatures.

The authors should replace the terminology “gender” by “sex” as they mean the normal biological difference between males and females.

Keyword

The referee thinks that the terminology “Clinicopathological” is not complete

 Introduction

- The references related to statistics should be recent ones, please update the first reference as it is since 2002.

Methodology

- The authors should assign the number of coauthor(s) and their initials who carried out the stages of the analysis staring from the initial screening till assessment of the bias.

- Did the authors check the references of the retrieved articles by manually screening? this issue was not clear in the text.

 Results

Tables 2 and 3: the study citation number should be mentioned with the name of the first author of each included study in the present analysis.

- It will be interesting if the authors run a “Trial sequential analysis (TSA)” to quantify the statistical reliability of the current meta-analysis significant findings (i.e. the results related to age and location) and confirm no needs for further future studies in this regard or vice versa.

 Minor comment

- Consent for publication is related to patients not the authors. In the current analysis, it is not applicable.
